# The Extended Information Systems Success Measurement Model: e-Learning Perspective

Teodora Vuckovic [1], Darko Stefanovic [1,*], Danijela Ciric Lalic [1], Rogério Dionisio [2], Ângela Oliveira [2] and Djordje Przulj [1]

1   Faculty of Technical Sciences, University of Novi Sad, 21000 Novi Sad, Serbia
2   DiSAC R&D Unit, Polytechnic Institute of Castelo Branco, 6000-084 Castelo Branco, Portugal
*   Correspondence: darko.stefanovic@uns.ac.rs

**Abstract:** This study investigated the crucial factors for measuring the success of the information system used in the e-learning process, considering the transformations in the work environment. This study was motivated by the changes caused by COVID-19 witnessed after the shift to fully online learning environments supported by e-learning systems, i.e., learning emphasized with information systems. Empirical research was conducted on a sample comprising teaching staff from two European universities: the University of Novi Sad, Faculty of Technical Sciences in Serbia and the Polytechnic Institute of Castelo Branco in Portugal. By synthesizing knowledge from review of the prior literature, supported by the findings of this study, the authors propose an Extended Information System Success Measurement Model—EISSMM. EISSMM underlines the importance of workforce agility, which includes the factors of proactivity, adaptability, and resistance to change, in the information system performance measurement model. The results of our research provide more extensive evidence and findings for scholars and practitioners that could support measuring information system success primarily in e-learning and other various contextual settings, highlighting the importance of people's responses to work environment changes.

**Keywords:** IS success factors; IS success measurement; UTAUT; workforce agility; COVID-19 changes; work environment changes; e-learning

## 1. Introduction

In searching for an information system (IS) success measure, there are nearly as many measures as there are studies. The reason for this is understandable when one considers that "information", as the output of an IS or the message in a communication system, can be measured at different levels, including the technical, semantic, and effectiveness levels [1]. The IS creates information that is communicated to the recipient, who is then influenced or not by it. The information flows through a series of stages—from production to use or consumption to affect individual and/or organizational performance. Hence, it is acknowledged that the IS's success needs to be expanded by recognizing what measures need to be undertaken to obtain the most suitable outputs. Accordingly, for more than a decade, many researchers have debated IS success [2–8].

Since the beginnings of research in this field, with the advent of the first model by DeLone and McLean (D&M IS Success Model) [1], several authors have confirmed the proposed model by examining the success of different ISs [8]. After the D&M model, the literature was amplified with studies that not only suggested and tested other IS success models [2–7], but also dealt with further development, modification, and adaptation of existing IS success models [9,10]. Finally, the authors of the Unified Theory of Acceptance and Use of Technology (UTAUT) [2] published the UTAUT model in 2003, which was created by combining the previous eight theories and models of IS success. Through years of using the UTAUT model in examining the performance of ISs in different environments, researchers identified its shortcomings and suggested improvements to the model.

Furthermore, there have been scientific contributions in the form of modified models for measuring the success of IS. For example, [11] found that many articles and studies cited the original UTAUT as a general reference for examining factors influencing the acceptance and use of technology. Still, researchers needed to deal with the expansion of UTAUT [11]. The literature review results show that there was some work on improving UTAUT. However, it only concerned improving existing model factors [11]. In 2012, authors [11] expanded the theory of the UTAUT, thus creating an Extended Unified Theory of Acceptance and Use of Technology (UTAUT2) [11] by adding three new factors that directly affect the behavioral intention to use the IS and the use of technology itself.

Based on the reports of the world's largest companies in the information technology industry [12,13], information and technological structure investments have taken almost the largest share of company investments. Given that technology investments are mainly based on implementing new or upgraded ISs, companies strive to foresee that these investments will pay off in future business. Therefore, it is necessary to determine the success of ISs implemented in the workplace. Different factors affect their success depending on the context in which the ISs are observed. Accordingly, these factors need to be measured appropriately. In line with the current state-of-the-art findings, an IS success model with factors that emphasize the importance of people's responses to changes in the work environment has yet to be found.

This study assessed the factors essential for measuring e-learning IS success when changes in the work environment happened. It was mainly motivated by the changes caused by COVID-19 witnessed after the shift to fully online learning environments, which forced the workforce to embrace technology.

Observing that e-learning was spotlighted much before, it is noteworthy that the true power of ISs in higher education was revealed by the disruptions and changes COVID-19 brought. To face uncertainties, higher education institutions required an agile, proactive, adaptive workforce with a positive attitude towards changes.

Over the past two decades, using learning management systems has attracted increasing interest from researchers around the globe. Previous research results brought us many different instruments and models for measuring IS success, its shortcomings and benefits in learning. However, it must be acknowledged that previous studies have yet to consider the impact of end-user agility on the widespread use and success of the system. COVID-19 did not ask; it disrupted the existing work environment, due to which the only survival pre-condition was fully transferring to IS and e-learning. Consequently, the authors of this paper sought to investigate how unpredictable changes and disruptions affect the learning environment and whether workforce agility affects the success of the IS in order to cover this gap.

Namely, empirical research was conducted on a sample comprising teaching staff from two European universities: 381 respondents from the University of Novi Sad (UNS), Faculty of Technical Sciences, Serbia and 149 respondents from the Polytechnic Institute of Castelo Branco (PICB), the School of Technology, Portugal.

IS and information and communication technology play a crucial role in developing agility related to speed and flexibility in responding to change [14]. Consequently, the results of this research provide a solid basis for enriching the previously tested and confirmed UTAUT model with factors that address workforce agility and its effect on overall IS success. In addition, the research results provide more extensive evidence and findings for scholars and practitioners that could support measuring IS success in various contextual settings, emphasizing the importance of people's responses to changes in the work environment.

Eventually, the main contribution of this research is the proposed Extended Information Systems Success Measurement Model (EISSMM), created by combining factors from the UTAUT model and the Theory of Workforce Agility, offering a road map for future research.

The rest of this paper is structured as follows. The upcoming section reviews the literature on the taxonomy of IS success and the role of workforce agility in IS success.

Section 3 presents the research framework and hypothesis, followed by an explanation of the research instrument, data collection process, and sample demographics in Section 4. In Section 5, statistical data and results are presented, and Section 6 discusses results, followed by a conclusion with limitations and directions for future research.

## 2. Background

### 2.1. Taxonomy of Information Systems Success

As seen throughout the previous research and discussions on IS success, many authors have argued over what a successful IS is. However, although much research has been conducted in this area, a single definition of IS success is still not found [15].

IS success research has approached this issue in various ways. Many studies have been conducted over the last decade and a half seeking to identify factors contributing to IS success. However, the dependent variables in these studies on IS success have yet to be discovered.

In recognition of this importance, this paper explores the research that has been conducted involving IS success since the first presentation of the IS success challenge, attempting to synthesize this research into a more coherent body of knowledge. It covers the last ten years and reviews all the empirical studies that have tried to measure some aspects of IS success.

Authors DeLone and McLean presented the D&M IS Success Model in 1992 [1] and thus opened a new chapter, inviting researchers to test the proposed model in different environments. Before they presented this model, researchers were less concerned with IS success, but after publicizing the D&M model, this area expanded. Ten years after, the same authors presented the Updated D&M IS Success Model, based on the previous decade of testing the initial model [16].

After D&M's traction, the Technology Acceptance Model (TAM) was developed in 2003. Davis presented the importance of factors instigating accepting or not accepting information technology. According to the TAM model's definition, the system's perceived usefulness and ease of use are an individual's two most essential expectations about using information technology. Based on the perceived ease of use and usefulness of the system, users develop an attitude towards use and intention to use, which affect the actual usage of the system. Davis believes that, viewed from the perspective of technology users, the perceived usefulness of a system is the strongest predictor of an individual's intention to use information technology [17].

To respond to the need to investigate further the items that make up the perceived usefulness of the system factor from the TAM model [17], bearing in mind that their effect changes over time as the user becomes more experienced in using the IS, an Extended Model of Technology Acceptance was created (TAM2) [18].

Besides the above-presented models, many theories have also been influential in this domain. The Theory of Planned Behavior [19], Social Cognitive Theory [20], and finally, the Unified Theory of Acceptance and Use of Technology gave additional frames to researchers. Based on a review of the extant literature, authors [3] developed UTAUT as a comprehensive synthesis of prior technology acceptance research. UTAUT compounded the essential factors of eight previous theories and models, opening different angles for observing and measuring IS acceptance and success.

As of today, several scientifically confirmed models and theories for measuring IS success exist. However, it is apparent that there needs to be a consensus on the measures of IS success.

Through years of testing models in different contexts and environments, researchers have found shortcomings and proposed improvements, and as contributions, have presented new models for measuring IS success. Examples of such models that have been empirically validated are TAM2, TAM3, and UTAUT2, and other variations exist. Table 1 contextualizes the distribution of previous research according to the models used.

**Table 1.** Distribution of previous research according to the models used.

| Model | Studies | No. Studies | % |
|---|---|---|---|
| D&M | [21–58] | 37 | 52 |
| TAM | [59] | 1 | 1 |
| Combined model | [60–90] | 31 | 43 |
| New model | [71,91,92] | 3 | 4 |

Most of the research was conducted using the D&M IS Success [1]. Around 50% of the papers were based on this single model. At the same time, 42% of papers proposed extensions of the basic model by introducing new, intermediate items and factors that can influence the success of the IS. Notably, nearly half of the conducted studies proposed a new model that combines items and factors from different models to overcome their limitations. Three papers proposed a completely new model for measuring information system success, two in the context of e-learning [91,92] and one in the context of e-government [71].

Reviewing IS success variables, it could be concluded that no single measure is intrinsically better. So, the choice of a success variable is often a function of the objective of the study—the organizational context, the aspect of IS that is addressed by the study, the independent variables under investigation, the research method, and the level of analysis, i.e., individual, organization, or society [1].

Developing improved measures for key theoretical constructs is a priority in this field. However, aside from their theoretical value, better measures for predicting and explaining system use would have great practical value, both for vendors who would like to assess user demand for new design ideas and IS managers within user organizations who would like to evaluate these vendor offerings [17].

The authors of the UTAUT model [2] state that several papers and articles use the original UTAUT as the initial reference for examining the acceptance and use of technology, but researchers still need to extend UTAUT. Their literature review shows that there was some work to improve UTAUT. However, this only refers to improvement of the model's existing factors [11]. Following the articles that utilized UTAUT, future research should focus on identifying constructs that can widen the prediction of intention and behavior over and above what is already known and understood.

Some studies have replicated TAM focused on the TAM model's psychometric aspects factors [93,94]. Other studies have supported the relative importance of TAM factors—perceived usefulness and perceived ease of use [95]. Finally, other studies have expanded TAM to include additional factors determining the technology acceptance model [18,95].

Despite the widespread use of subjective measures in practice, little has been paid to the measures' quality or how well they correlate with user behavior. By discovering the low usage correlations often observed in research studies, it is noteworthy to avoid business decisions based on unvalidated measures and consider the need to be more informed about users' system acceptance [17,96].

While ISs are critical resources for an organization [97], the people using these systems and the information derived from them can influence the system's resulting success.

### 2.2. The Role of Workforce Agility in Is Success

Workforce agility is defined as the employee's speed of action and flexibility for change [14]. The term agile workforce implies that employees proactively innovate and develop their skills, even before the immediate requirement to create such skills [98,99]. The agile workforce is said to move flexibly, quickly, and efficiently in any work environment [100].

Workforce agility is one company performance driver since it enables companies to adapt quickly to a constantly changing business environment. Based on workforce agility, companies can adapt to changes and thrive in new environments because their employees can process information quickly and proactively and even take initiatives for self-improvement [36]. Accordingly, in many studies, an agile workforce was found to be

essential for the project's success [101]. Therefore, the quality of the above performance largely depends on workforce competence [35,67–72].

New or upgraded ISs are often made available to employees to help them perform their tasks to achieve shorter development periods, decentralization, flexibility, customization, and resource efficiency [33]. Therefore, workforce agility is crucial for IS success.

### 3. Research Framework and Hypothesis Development

*3.1. Research Framework: The Extended Information Systems Success Measurement Model (EISSMM)*

This paper explores the research that has been conducted involving IS success measurement as an essential phase of the whole IS lifecycle. We attempted to collect empirical studies that measured some aspects of IS success. Taken together, these 145 references provide a representative review of the work performed and provide the basis for formulating a more comprehensive model of IS success than has been attempted in the past.

UTAUT is one of the most often-used theories in testing the success and acceptance of technical innovation or information systems [6,102–104]. The introduction of information technology in the work process is the most common type of change in the work environment. Such a transformation requires a response from employees in the direction of technology acceptance [14,19,99,100,105–108]. Employees resist change for various reasons—fear of the new, ignorance of information technology, etc. Thereby, an important thing to consider is the characteristics of employees that affect the response and acceptance of a change in the usual way of working.

Evaluating all the knowledge gathered by reviewing previous research results in the literature, the authors of this article propose the Extended Information System Success Measurement Model—EISSMM. This model stresses the importance of including an additional concept—workforce agility—in the IS performance measurement model. The EISSMM research model is shown in Figure 1.

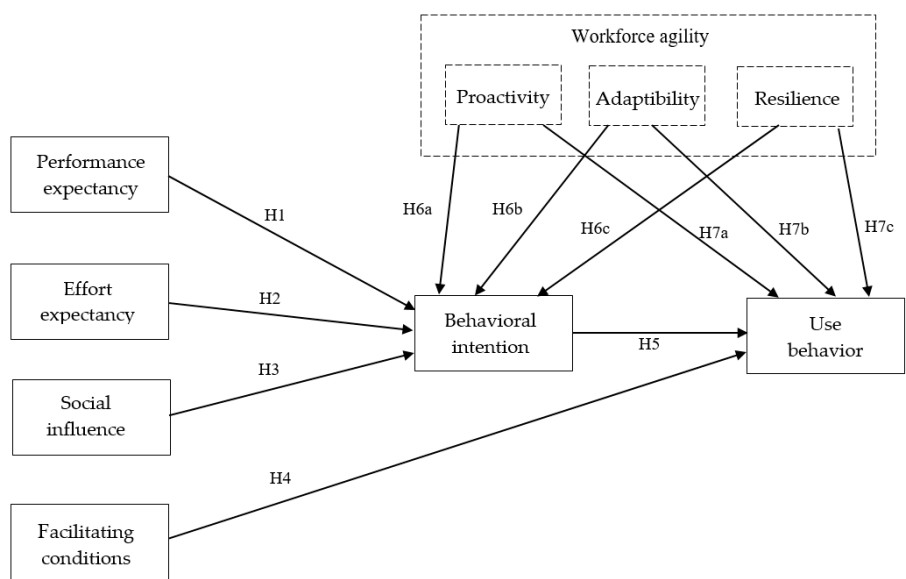

**Figure 1.** EISSMM research model.

*3.2. Hypotheses Development*

This research model includes six factors of the UTAUT model where: (1) performance expectancy is the level to which an individual believes that using an IS will increase their work performance and ensure easier progress, (2) effort expectancy is the level of ease of use of the system from the user's perspective, (3) social influence is the level to which a person perceives the importance of the opinion of other people from the immediate environment about using the IS, (4) facilitating conditions is the level to which a person believes in the

existence of organizational and technical infrastructure to support the use of the IS, (5) behavioral intention is a person's overall reaction to using the system, (6) use behavior refers to the activities through which a person uses the IS [2]. The three factors of workforce agility we included were those where: (1) proactivity refers to situations when a person initiates activities that have positive effects on the changed environment [105,107], (2) adaptability is based on a person's ability to change or adjust themselves and their behavior to better fit into a new environment [105], and (3) resilience reflects a person's ability to function effectively under stress, despite a changing environment or in situations where coping strategies have failed. This factor describes personality traits: positive attitude towards changes, new ideas, and technology; tolerance for uncertain and unexpected situations as well as differences in opinions and approaches; and tolerance for stressful situations and coping with stress [107]. The behavioral intention and use behavior factors are the dependent variables in the proposed model.

Performance Expectancy → Behavioral Intention Relation

Through the performance expectancy factor, information system users state their expectations about the usefulness of the IS, whether the use of the IS will increase their work efficiency and productivity, and whether it will facilitate their entire work process. With all the items that comprise the performance expectancy factor, it is assumed that it affects the intention of users to use the IS (behavioral intention).

The relationship between performance expectancy and behavioral intention to use the system has been observed and examined in various contexts of IS success research. From IS acceptance in the e-learning context [4,5,109–111] to online banking [103], social networks in data exchange processes [105], employee portal success [112], etc., expected performance has been shown as a factor that has a significant and direct impact on the behavioral intention to use an IS. Therefore, the first hypothesis, adopted from the UTAUT model, is:

**Hypothesis 1 (H1).** *Performance expectancy positively affects end user behavioral intention to use the system.*

Effort Expectancy → Behavioral Intention Relation

Regarding effort expectancy and behavioral intention to use the system, the vast majority of previous research has shown the existence and significance of the relation between the two factors [4,5,109,110,112–114]. According to the theory of the UTAUT model, the expectancy of the effort that needs to be invested in using the IS directly influences the user's behavioral intention to use that IS. Therefore, the second hypothesis, adopted from the UTAUT model, is:

**Hypothesis 2 (H2).** *Effort expectancy positively affects end user behavioral intention to use the system.*

Social Influence → Behavioral Intention Relation

The UTAUT model authors claim that the influence of colleagues from the person's team, the institution where the person works, and even the influence of the broader institution itself play a behavioral role in the intention of users to use the IS. As the social influence factor is a significant predictor of behavioral intention to use the IS, many previous types of research have examined this relationship. When looking at the factors influencing students' intention to use IS for e-learning, social influence was not evidenced to be a significant determinant of behavioral intention [5,111]. Additionally, this relationship still needs to be confirmed in adopting social networks for data sharing [104]. However, all other studies have shown social influence on the behavioral intention to use the IS, thus confirming the existence of this relationship [3,4,109,110,112–116]. By analyzing previous empirical research, the authors defined the third hypothesis:

**Hypothesis 3 (H3).** *Social influence positively affects end user behavioral intention to use the system.*

Facilitating Condition → Use Behavior Relation

Facilitating conditions represent the subjective assessment of IS users on the organizational and technical infrastructure supporting the system's use. Even though, according to the original UTAUT model, the facilitating conditions factor significantly influences the use behavior of the IS, previous research results on the existence of this relation are divided. On the one hand, several studies confirmed the positive effect of facilitating conditions on use behavior in the context of e-learning [4,5,110,115,116]. On the other hand, also in the context of e-learning, this relationship was not confirmed [3,6,104,109,113,117]. Even though previous research results on this relation have been diverse, taking into account the original UTAUT model and the results that evidence the importance of this relation, the authors defined the fourth hypothesis as follows:

**Hypothesis 4 (H4).** *Facilitating conditions positively affect use behavior.*

Behavioral Intention → Use Behavior Relation

The behavioral intention factor, which addresses the assumed temporal determinant of system use according to the UTAUT, significantly impacts ultimate behavior in using the system. The existence of this relation was confirmed in many previous studies. In the context of IS success in e-learning, internet banking, knowledge management systems, etc. [4,5,104,110,113,115,116], behavioral intention has been evidenced to be a significant determinant of use behavior. Therefore, the fifth hypothesis is defined as follows:

**Hypothesis 5 (H5).** *End user behavioral intention to use the system positively affects use behavior.*

Beyond the six factors from the UTAUT model, workforce agility is a component that affects how the IS will be used when brings changes to the work environment and the usual way of working [118]. Previous research on workforce agility only examined relations of proactivity, adaptability, and resilience as dependent factors [105,107,108,118–120]. According to the results, no research was found in the literature examining the impact of workforce agility on the behavioral intention to use the IS and the use behavior. Therefore, to answer the research question, does the workforce's agility affect the success of IS? the authors constructed the hypotheses explained below that would help measure IS success more accurately.

Workforce Agility → Behavioral Intention Relation

Minnesota's Theory of Changes in the Work Environment explains "work" as an interaction between the individual and the work environment. At the same time, adaptation to changes in work reflects the mutual action of one and the other to meet joint needs [106]. The workforce agility construct explains people's behavior in situations where changes occur. Introducing, replacing, or upgrading ISs as a form of innovation in the work environment is a persistent change in work processes. For this reason, it is important to examine whether and to what extent workforce agility affects the behavioral intention of users to use the IS. According to the above, the authors proposed the sixth hypothesis:

**Hypothesis 6 (H6).** *Workforce agility positively affects end user behavioral intention to use the system.*

Workforce agility is built by proactiveness, adaptability, and resilience [107]. Therefore, the authors defined three auxiliary hypotheses:

**Hypothesis 6a (H6a).** *Proactivity positively affects end user behavioral intention to use the system.*

**Hypothesis 6b (H6b).** *Adaptability positively affects end user behavioral intention to use the system.*

**Hypothesis 6c (H6c).** *Resilience positively affects end user behavioral intention to use the system.*

Workforce Agility → Use Behavior Relation

Matching or adapting employees' skills to new requirements imposed by the work environment affects employee satisfaction. Satisfaction arises from matching the employee's needs and values with the work environment's requirements. Satisfaction and a sense of achievement ultimately contribute to work quality and performance [106]. Therefore, the more agile the workforce is in changing conditions, the more their agility contributes to better adaptation and a greater willingness to use IS and new technologies.

According to the above, the authors assumed that workforce agility has an impact on use behavior, and thus proposed the seventh hypothesis with three auxiliary hypotheses listed below:

**Hypothesis 7 (H7).** *Workforce agility positively affects use behavior.*

**Hypothesis 7a (H7a).** *Proactivity positively affects use behavior.*

**Hypothesis 7b (H7b).** *Adaptability positively affects use behavior.*

**Hypothesis 7c (H7c).** *Resilience positively affects use behavior.*

## 4. Methods and Materials

### 4.1. Research Instrument

This research followed a quantitative approach, being implemented through the survey method.

According to the analyzed theoretical models and reviewing the relevant literature sources, the authors operationalized nine factors (6 representing IS success and 3 representing workforce agility) with 48 manifest variables in the final questionnaire. Per guidelines for theoretical models, 39 out of 48 manifest variables were grounded, and depending on the specific functionalities of the IS, the rest varied. Given that the use behavior factor depends entirely on the functionality of the IS under analysis, nine variables representing this factor were identified to examine the IS success in the teaching process for this research. The final model with factors and associated manifest variables is presented in Table 2.

This research used a self-reporting (subjective) assessment of IS success and workforce agility as perceived by respondents. To capture respondents' subjective assessment of performance expectancy, effort expectancy, social influence, and facilitating conditions, a continuum of five-point, unipolar Likert-type scale from 1,"strongly disagree", to 5, "completely agree", was used. A five-point, unipolar Likert-type scale from 1, "never", to 5, "very often", was used to capture respondents' subjective assessment of behavioral intention and use behavior. Ten system functionalities were observed through the use behavior factor within this study: announcements, discussion forum, learning materials, video materials, assignments, instructions, test, chat, gradebook, and participants list.

**Table 2.** Nine factors with associated manifest variables.

| Factor | Manifest Variable | Sources |
|---|---|---|
| Performance expectancy (PE) | Usage in the working process | [2,121] |
| | Faster obligation fulfillment | [2,121] |
| | Increase in work productivity | [2,121] |
| | Easier working | [2] |
| | Better learning performance | [2] |
| Effort expectancy (EE) | System usage: clear and understandable | [2,121] |
| | Fast system understanding | [2,121] |
| | Simplicity of using | [2,121] |
| | Learning to handle the system easily | [2,121] |
| | System responsiveness | [2] |
| Social influence (SI) | Colleague influence | [2,121] |
| | Team co-worker influence | [2,121] |
| | Colleagues' willingness to help influence | [2,121] |
| | Institution influence | [2,121] |
| | The feeling of belonging | [2] |
| Facilitating conditions (FC) | Owning resources | [2] |
| | Owning competence | [2] |
| | Compatibility with other systems | [2] |
| | Fitting into the way of working | [2] |
| | User manual instructions | [2] |
| Behavioral intention (BI) | Intention to use the system in the future | [2,121] |
| | Prediction of future usage | [2,121] |
| | Planning to use the system in the future | [2,121] |
| Use behavior (UB) | System functionalities * | [122] |
| Proactivity (P) | Seek work improvement opportunities | [107,120,123] |
| | Seek effective ways to work | [107,119,120,123] |
| | Leaving it to chance; not reacting | [105,107] |
| | Adherence to work rules and procedures | [105,107] |
| | Finding additional resources at work | [119,120,123] |
| Adaptability (A) | Adaptive to team changes | [119,120,123] |
| | Critical feedback acceptance | [105,107] |
| | Adaptive to the new situation | [107,120,123] |
| | New equipment use | [119,123] |
| | Keeping up to date | [119,123] |
| | Adaptive to tasks switching | [119,123] |
| Resilience (R) | Efficiency in stressful situations | [107,119,120,123] |
| | Working under pressure | [107,119,120,123] |
| | Reaction to problems | [107] |
| | Taking action | [119,123] |

* Use behavior factor corresponds to specific functionalities of the observed IS, so the number of manifest variables that build it varies.

### 4.2. Sample and Data Collection

The sample used in this research was non-probabilistic, and individuals were selected using expert sampling as a type of purposive sampling technique. The sample consisted of the teaching staff from two European universities: the University of Novi Sad, Faculty of Technical Sciences in Serbia, and the Polytechnic Institute of Castelo Branco, the School of Technology in Portugal.

Due to COVID-19, which has accelerated the shift to fully online learning environments, the teaching staff of both universities faced a change that implied a complete transition from classic classroom studies to distance learning through e-learning ISs, forcing them to embrace technology. For conducting face validity, the final version of the questionnaire was tested. A pilot survey with 19 respondents was initially performed. After completing the pilot study, certain questionnaire items were corrected to clarify them in the final survey. Respondents' subjective perceptions of using the IS in the teaching process were assessed by distributing the questionnaire electronically through the online data collection tool SurveyMonkey. Invitations to participate in the research were sent electronically according to the researchers' recommendations [124] and in a pre-defined

order. Participation invitations in the study and a link to the electronic questionnaire were sent to the teaching staff. Filling out the questionnaire and participating in the research were voluntary. Therefore, none of the participants was forced to answer in any way.

The e-learning IS at the UNS has a total of 752 registered users, of which 462 accessed the link with the questionnaire, while 403 users filled out the entire questionnaire. Accordingly, the response rate was 53.6%. To ensure the validity of the research results, incomplete answers were omitted through the initial data screening procedure. By calculating the standard deviation of each respondent's answer, 22 responses were removed from further analysis, resulting in the final sample of 381 from UNS used in the study.

Moreover, the e-learning IS at the PICB has 354 registered users, of which 202 accessed the link with the questionnaire, and 161 users filled out the entire questionnaire. Accordingly, the response rate was 45.5%. After initial data cleaning, incomplete responses were omitted through a non-inclusion bias interpretation procedure. Finally, 12 responses were removed from further analysis, resulting in the final sample of 149 from the PICB used in this study. The demographic characteristics of the respondents from the UNS and the PICB are explained in detail below.

### 4.3. Sample Demographics

The sample consisted of respondents across different categories of gender, age, and academic titles. The 381 respondents from the UNS encompassed staff with other academic titles, including 26 (6.8%) teaching associates, 107 (28.1%) teaching assistants, 98 (25.7%) assistant professors, 86 (22.6%) associated professors, 44 (11.5%) full professors, and 20 (5.2%) self-declared others, while 149 respondents from the PICB held the following titles: 12 (8.1%) professor coordinators, 84 (56.4%) professors, 4 (2.7%) assistants, 29 (19.5%) invited assistants, 17 (11.4%) guest lecturers, and 3 (2%) declared as others.

The largest share of UNS respondents was in the second category—respondents between 31 and 40. The smallest percentage of respondents fell into the last, fifth category—over 60 years old. At the PICB, most respondents were in the third category—between 41 and 50—and the fourth category—between 51 and 60 years old. The smallest percentage of respondents fell into the last, fifth category—over 60 years old. To obtain more reliable information about e-learning IS success, it was necessary to determine the respondents' previous experience using this or similar ISs, and the average e-learning IS daily usage. The distribution of respondents according to their expertise and everyday usage is shown in Table 3.

**Table 3.** Sample structure according to experience in using IS and average daily usage.

|  | No. | % | No. | % |
|---|---|---|---|---|
| **Experience in using e-learning IS** | **UNS** | | **PICB** | |
| Without prior experience | 140 | 36.7 | 10 | 6.7 |
| Less than 1 year | 64 | 16.8 | 38 | 25.5 |
| Between 1 and 3 years | 62 | 16.3 | 14 | 9.4 |
| More than 3 years | 115 | 30.2 | 87 | 58.4 |
| Total | 381 | 100.0 | 149 | 100.0 |
| **Average e-learning IS daily usage** | | | | |
| Less than 1 h | 223 | 58.5 | 29 | 19.5 |
| Between 1 h to 3 h | 140 | 36.7 | 74 | 49.7 |
| Between 3 h to 5 h | 15 | 3.9 | 34 | 22.8 |
| Between 5 h to 7 h | 3 | 0.8 | 11 | 7.4 |
| More than 7 h | 0 | 0 | 1 | 0.7 |
| **Total** | 381 | 100.0 | 149 | 100.0 |

## 5. Results

First, exploratory factor analysis (EFA) was conducted to identify the structure of factors by examining the correlation matrices. For EFA, the maximum likelihood method of extraction was used. After determining the structure of the factors, confirmatory factor analysis (CFA) was performed to verify the factor structure of a set of observed manifest variables. The next was to calculate the instrument's reliability through the Cronbach alpha coefficient, $\alpha$, which shows the degree to which the instrument is "free" from measurement error and represents the ratio of the sample variance to the total variance of the instrument [125]. Furthermore, to examine the reliability of the complete instrument, a confirmatory factor analysis was conducted using exploratory structural equation modeling (ESEM) [126]. Finally, discriminant and convergent validity were calculated for all factors: composite reliability (CR), average variance extracted (AVE), maximum shared variance (MSV), and average shared variance (ASV). In the end, structural equation modeling (SEM) was performed to determine the significance of the relationships between the factors. Based on the goodness-of-fit index, the final rating of the SEM model was determined as in the CFA.

Additionally, the variances of each factor in the model were analyzed by calculating and observing the squared multiple correlations ($R^2$), which represent the percentage of explained variance for the observed factor. The higher the $R^2$ percentage, the higher the predictive power of the assumed model [127]. In addition, path coefficients between factors in the model (path coefficients—$\beta$) were calculated, representing the importance of relationships between factors. The IBM SPSS tool for statistical data processing was used. The results of the aforementioned statistical analyses for the UNS and the PICB are presented in the following section.

### 5.1. UNS Results

EFA: The extraction results yielded a nine-factor solution for the eigenvalues greater than one. The values of the scree plot also confirmed the nine-factor solution. Moreover, the nine-factor solution yielded a good result in the percent of the cumulative sample variance (71.66%). Factor analysis was conducted iteratively until an adequate model and factor structure that satisfied all the criteria were achieved. Among 48 items, 11 items had shallow factor loading scores. Thus, these items were omitted from the matrix structure because they built other constructs that were not important for this research. Hence, the final model with nine factors and 37 items/manifest variables was accepted.

CFA: After omitting items with deficient factor loading scores and accepting the final factor structure, confirmatory factor analysis was conducted to test the reflective model and statistically confirm the factors obtained via EFA. The model with nine factors and 37 items had adequate model fit indices according to the recommended values. With a significance level of n < 0.05 for the Chi-square and the adequacy of all the suitability index values, it can be concluded that the measurement model fully describes the obtained data. According to these values, the measurement model has adequate goodness.

Reliability and validity assessment: The calculated Cronbach's alpha values for each dimension were performance expectancy = 0.949, effort expectancy = 0.913, social influence = 0.701, facilitating conditions = 0.732, behavioral intention = 0.978, use behavior = 0.741, proactivity = 0.810, adaptability = 0.861, and resilience = 0.828. According to Hair [128], the minimum criterion for each dimension to be valid is 0.60, leading to the conclusion that all our dimensions satisfy the aforementioned criteria. The CR, AVE, MSV, and ASV coefficient values for all factors are shown in Table 4. Considering the calculated coefficient values, it can be concluded that the measurement model has adequate reliability, convergent validity, and discriminant validity.

**Table 4.** UNS—Discriminant and convergent validity of the measuring instrument.

|  | CR | AVE | MSV | ASV | BI | PE | A | EE | FR | R | UB | SI | P |
|---|---|---|---|---|---|---|---|---|---|---|---|---|---|
| BI | 0.979 | 0.938 | 0.276 | 0.129 | 0.969 [a] | | | | | | | | |
| PE | 0.949 | 0.789 | 0.425 | 0.142 | 0.456 | 0.888 [a] | | | | | | | |
| A | 0.863 | 0.562 | 0.530 | 0.194 | 0.308 | 0.355 | 0.750 [a] | | | | | | |
| EE | 0.916 | 0.685 | 0.425 | 0.162 | 0.333 | 0.652 | 0.437 | 0.828 [a] | | | | | |
| FR | 0.761 | 0.532 | 0.179 | 0.103 | 0.307 | 0.137 | 0.406 | 0.423 | 0.729 [a] | | | | |
| R | 0.814 | 0.595 | 0.530 | 0.180 | 0.343 | 0.317 | 0.728 | 0.395 | 0.373 | 0.772 [a] | | | |
| UB | 0.737 | 0.588 | 0.105 | 0.059 | 0.324 | 0.233 | 0.314 | 0.153 | 0.078 | 0.314 | 0.698 [a] | | |
| SI | 0.803 | 0.607 | 0.276 | 0.121 | 0.525 | 0.381 | 0.295 | 0.361 | 0.368 | 0.323 | 0.262 | 0.779 [a] | |
| P | 0.840 | 0.646 | 0.257 | 0.093 | 0.171 | 0.244 | 0.507 | 0.283 | 0.296 | 0.439 | 0.129 | 0.154 | 0.804 [a] |

[a] The values in the diagonal are the square root of the AVE.

SEM: Fit indices for SEM show that all values were acceptable, indicating excellent model fit ($\chi^2/df$ = 1.752; NFI = 0.916; CFI = 0.962 and RMSEA = 0.044). Figure 2 shows the SEM model with question coefficient values ($\beta$), t-values, and multiple correlation squares ($R^2$).

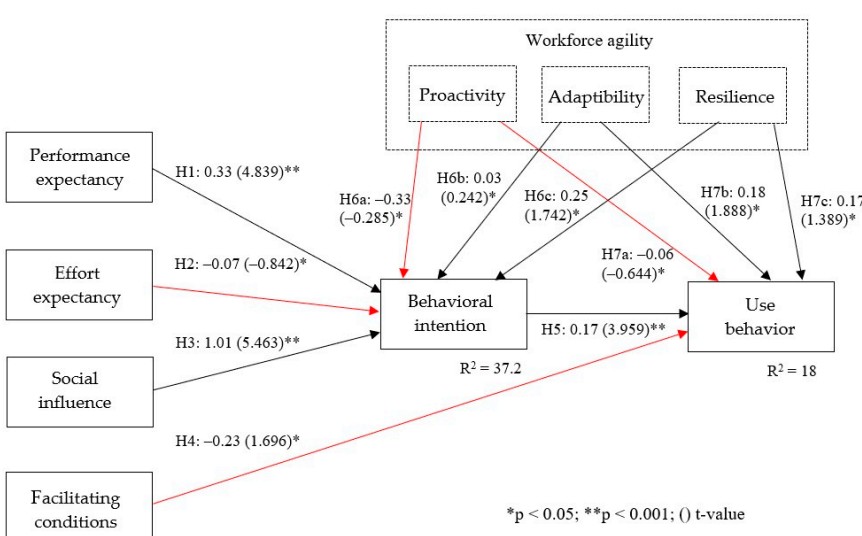

**Figure 2.** UNS—SEM model.

The values of the path coefficients resulted in acceptance, i.e., confirming or rejecting the assumed relationships between the factors in the model. The relationship between the performance expectancy and the behavioral intention to use the system is statistically significant and positive, as evidenced by the path coefficient $\beta$ = 0.33; t-value = 4.839. The relationship between the effort expectancy and behavioral intention factors is negative and statistically insignificant according to the respondents' answers ($\beta$ = −0.07; t-value = −0.842). The social influence factor has a statistically significant influence on the behavioral intention factor, with an above-average value of the path coefficient $\beta$ = 1.01; t= 5.463. This relationship in the model is the strongest and statistically the most significant. The relationship between the facilitating conditions and use behavior factors is statistically insignificant and not confirmed ($\beta$ = −0.23; t-value = 1.696).

The relationship examining whether the behavioral intention factor positively affects the use behavior factor was confirmed and is statistically significant ($\beta$ = 0.17; t-value = 3.959). The relations between the workforce agility construct and behavioral intention and use behavior factors were examined for each factor that constructs workforce agility. Thus, the relationship between proactivity and the behavioral intention was statistically

insignificant. This negative relationship is confirmed by the path coefficient $\beta = -0.33$ and t-value = $-0.285$. In addition, proactivity shows a negative relationship with the use behavior factor ($\beta = -0.06$; t-value = $-0.644$). Factor adaptability resulted in a positive and statistically significant impact on both behavioral intention ($\beta = 0.03$; t-value = $0.242$) and use behavior ($\beta = 0.18$; t-value = $1.888$), whereas the relation of adaptability to use behavior was stronger. The relation between resilience and behavioral intention is positive and statistically significant ($\beta = 0.25$; t-value = $1.742$). Additionally, a significant relationship between resilience and use behavior is confirmed ($\beta = 0.17$; t-value = $1.389$). Finally, the positive impact of workforce agility on both behavioral intention and use behavior is evidenced.

### 5.2. PICB Results

EFA: The extraction results admitted an eight-factor solution for the eigenvalues greater than one. Due to the low loadings, the facilitating conditions factor with accompanied manifest variables was eliminated at that stage. The values of the scree plot also confirmed the eight-factor solution. Moreover, the eight-factor solution yielded a good result in the percent of the cumulative sample variance (68.97%). Factor analysis was conducted iteratively until an adequate model and factor structure that satisfies all the criteria were achieved. Among the remaining variables, eleven had low factor loading scores. Thus, these items were omitted from the matrix structure because they built other constructs that are not relevant to this research. Hence, the final model with eight factors and 30 items/manifest variables was accepted.

CFA: After omitting items with shallow factor loading scores and accepting the final factor structure, CFA was conducted to test the reflective model and statistically confirm the factors obtained via EFA. The model with eight factors and 30 manifest variables had adequate model fit indices according to the recommended values. With a significance level of n < 0.01 for the Chi-square and the adequacy of all the suitability index values, it can be concluded that the measurement model fully describes the obtained data. According to these values, the measurement model has adequate goodness.

Reliability and validity assessment: Calculated Cronbach's alpha values for each dimension were performance expectancy = 0.881, effort expectancy = 0.885, social influence = 0.720, behavioral intention = 0.957, use behavior = 0.806, proactivity = 0.558, adaptability = 0.792, and resilience = 0.833. According to Hair [128], the minimum criterion for each dimension to be valid is 0.60, leading to the conclusion that all our dimensions satisfy the abovementioned criteria. The CR, AVE, MSV, and ASV coefficient values for all factors are shown in Table 5. Considering the calculated coefficient values, it could be concluded that the measurement model has adequate reliability, convergent validity, and discriminant validity.

**Table 5.** PICB—Discriminant and convergent validity of the measuring instrument.

|      | CR    | AVE   | MSV   | ASV   | P       | PE      | EE      | A       | SI      | SU      | BI      | R       |
|------|-------|-------|-------|-------|---------|---------|---------|---------|---------|---------|---------|---------|
| P    | 0.817 | 0.619 | 0.238 | 0.097 | 0.787 [a] |         |         |         |         |         |         |         |
| PE   | 0.894 | 0.630 | 0.407 | 0.163 | 0.170   | 0.794 [a] |         |         |         |         |         |         |
| EE   | 0.883 | 0.655 | 0.407 | 0.210 | 0.274   | 0.638   | 0.809 [a] |         |         |         |         |         |
| A    | 0.756 | 0.620 | 0.367 | 0.183 | 0.488   | 0.254   | 0.428   | 0.787 [a] |         |         |         |         |
| SI   | 0.844 | 0.664 | 0.062 | 0.025 | 0.000   | 0.228   | 0.248   | 0.130   | 0.815 [a] |         |         |         |
| SU   | 0.717 | 0.462 | 0.279 | 0.125 | 0.246   | 0.290   | 0.343   | 0.377   | 0.106   | 0.680 [a] |         |         |
| BI   | 0.958 | 0.884 | 0.348 | 0.226 | 0.344   | 0.590   | 0.546   | 0.517   | 0.154   | 0.528   | 0.940 [a] |         |
| R    | 0.852 | 0.661 | 0.367 | 0.207 | 0.397   | 0.393   | 0.570   | 0.606   | 0.114   | 0.425   | 0.498   | 0.813 [a] |

[a] The values in the diagonal are the square root of the AVE.

SEM: Fit indices for SEM show that all values are acceptable, indicating excellent model fit ($\chi^2/df = 1.278$; NFI = 0.87; CFI = 0.97 and RMSEA = 0.043). Figure 3 shows the

SEM model with question coefficient values (β), t-values, and multiple correlation squares ($R^2$).

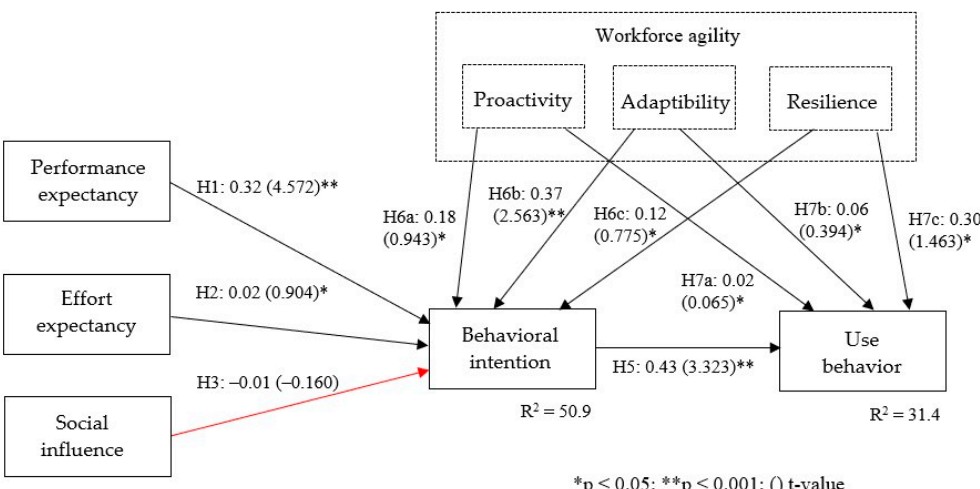

**Figure 3.** PICB—SEM model.

The relation between the expected performance and the behavioral intention is one of the most significant in the model, as evidenced by the path coefficient (β = 0.32; t-value = 4.572). The relation between the effort expectancy and behavioral intention factors is positive and statistically significant (β = 0.09; t-value = 0.904).

Social influence was statistically insignificant when observing its influence on the dependent behavioral intention factor, with a negative value of the path coefficient β = −0.01; t = −0.160. The dependent behavioral intention factor positively affects the second dependent factor, use behavior. This relation is statistically significant and most substantial in the model (β = 0.43; t-value = 3.323).

The relations between the workforce agility construct and behavioral intention and use behavior factors were examined for each factor that constructs workforce agility. Thus, the contested relationship between proactivity and behavioral intention is statistically significant, confirmed by the path coefficient β = 0.18; t-value = 0.943.

In addition, the proactivity shows a positive and statistically significant relationship with the use behavior factor (β = 0.12; t-value = 0.755). Adaptability resulted in a positive and statistically significant impact on behavioral intention (β = 0.37; t-value = 2.563) and use behavior (β = 0.06; t-value = 0.394) factors. The relation between resilience and behavioral intention is positive and statistically significant (β = 0.12; t-value = 0.755). Additionally, an important relationship between resilience and use behavior was confirmed (β = 0.30; t-value = 1.463). Finally, the positive impact of workforce agility on both behavioral intention and use behavior is evidenced.

## 6. Discussion

Beholding that COVID-19 has, so far, changed every traditional way of processing activities in the past two years [129–133], higher education institutions also seek this change to keep the learning process ongoing [134–144]. Therefore, online learning has become the new normal. However, the question of this transition's success arose, highlighting the need to understand what factors are crucial across the needed change. The most accurate software, i.e., ISs, have enabled the digital transformation of the learning environment, but have the teachers adapted to the new way of working? That is the question we found essential when measuring IS success.

In the vast e-learning IS success literature, no previous research observed the influence explained above. Consequently, this study investigated the potential of combining two widely accepted theories, shaped in a manner to fulfill the research gap and widen the IS success horizon.

The conceptual model was hypothesized with seven main and six auxiliary hypotheses in two region-different higher education institutions. The results confirmed five main and four auxiliary hypotheses for the UNS, while for the PICB, five primary and six auxiliary hypotheses were confirmed. The main contribution of this article was to test the given instrument in diverse countries, emphasizing the quest for workforce agility toward IS success in various learning environments.

The UTAUT model has been widely recognized and utilized in previous IS success and acceptance testing [3–5,109,110,112–116]. Our research found that performance expectancy was still one of the most decisive behavioral intention predictors [4–6,103,104,109–116] in both observed institutions. Moreover, effort expectancy was widely shown as essential for behavioral intention [4–6,103,104,109–116]; however, with $\beta = -0.07$ for the UNS, it was shown negative. At the UNS, 63% of respondents had experience using such an IS for a year or more, so it could be assumed that this is why the expected effort in using the IS did not influence the intention to use the system.

Although social influence was the strongest predictor of behavioral intention in the UNS, this relationship resulted in a hostile and statistically insignificant influence $\beta = -0.01$ at the PICB.

In the previously observed studies, the facilitating conditions towards the use behavior path coefficient ranged from $-0.23$ to $0.40$ [3–6,103,104,109–113,115–117]. Within this study, the facilitating conditions factor negatively influenced use behavior in both observed institutions. A possible explanation for these results is that the respondents needed to consider the facilitating conditions in this IS exclusive because they started using it as an addition to the existing digital work environment and the e-services they already use. In this regard, respondents are very likely to perceive facilitating conditions before using the given IS.

The relation between behavioral intention to use the system and use behavior was found to be statistically significant and positive in almost 90% of the previous studies [3,4, 6,103,110–112,115,117], which was also found in this study.

Finally, to evidence the importance of workforce agility's influence on the overall IS success and overcome research shortcomings, this paper's authors hypothesized WA with two main hypotheses. Seeking more extensive outputs on two main WA hypotheses, an additional six were set for assessing each WA aspect (proactivity, adaptability, and resilience).

Observing the first quested factor, behavioral intention, its dependent factor proactivity demonstrated a statistically non-significant influence on it at UNS ($\beta = -0.33$). However, at PICB, it was found to be strong and statistically significant ($\beta = 0.18$; $p < 0.05$). In the context of the environment in which the proposed model and all associated hypotheses were tested and based on other statistical analyses that demonstrated the model's significance and strength, it is possible to draw the following conclusion. Likewise, in hypothesis H4, respondents most likely do not associate proactivity characteristics only with the observed IS. In addition, the results show respondents significantly consider proactivity as an indicator of the overall success of the IS, being used as a result of a specific change. Factor adaptability positively predicted behavioral intention to use the system at the UNS and the PICB. The significance of this relationship at UNS was evidenced with a path coefficient $\beta = 0.03$ and a significance level of $p < 0.05$. At the PICB, the strength and significance were even higher ($\beta = 0.37$; $p < 0.01$ ). Therefore, the relationship between adaptability and behavioral intention was vindicated according to both auxiliary hypotheses. The relationship between resilience and behavioral intention to use the system was statistically significant and positive. At UNS, the significance of this relationship was evidenced with a path coefficient $\beta = 0.25$, $p < 0.05$, and at the PICB with $\beta = 0.12$, $p < 0.05$. The resilience factor is evidenced to significantly predict behavioral intention to use the system in both cases.

After analyzing the auxiliary relations among all three workforce agility constituents with the behavioral intention to use the IS in the teaching process at the UNS and the PICB, we conclude this relationship is confirmed and vital for IS success.

The second quested factor from this study, use behavior, was observed through WA influence. The results show that for the IS used in the teaching process at UNS, the impact of proactivity on the use behavior is not statistically significant ($\beta = -0.33$). However, in another case at the PICB, this relationship was found to be statistically significant ($\beta = 0.02$; $p < 0.05$). The influence of adaptability on use behavior was evidenced in both settings. At the UNS, adaptability was the most powerful of all three workforce agility constituents concerning use behavior ($\beta = 0.18$; $p < 0.05$). The last WA component, resilience, demonstrated a positive impact on use behavior in both questioned institutions.

Finalizing findings from all observed auxiliary hypotheses tested to question the importance of workforce agility in IS success, the authors of this paper demonstrated its significance. The proposed EISSMM model, combining essential factors from UTAUT and WA, can be used as an IS success measurement instrument in an e-learning context.

## 7. Conclusions

As we witness the change imposed by the pandemic in 2020, the usual teaching process has been entirely transferred to the digital environment. This change affected not only the teaching process but also all other processes regularly carried out on the physical premises of institutions. Changes in the working environment nowadays occur briefly and are often caused by introducing new, upgrading, or replacing existing ISs. Therefore, from the aspect of IS success and acceptance, it is essential to see how users react to these changes. Thus, the authors of this article expand the existing theory on the UTAUT model [2] with the construct of workforce agility [106,120].

As other IS success measurement models are based on examining the technical components of the system [1,16] and considering that the human factor has a decisive influence on the acceptance of technology, the authors decided to apply the UTAUT model. Namely, UTAUT starts from the assumption that the user's expectations about how they will use the system, whether they will have all the necessary instructions, and the perception of IS usefulness are critical in measuring the success and acceptance of the technology [11]. Additionally, the factor of workforce agility, through proactivity, adaptability, and resilience, illustrates the behavior of users in the work environment when a change occurs. These factors, together, form the Extended Information Systems Success Measurement Model. The EISSMM was conceived assuming it is independent of the context in which ISs are used.

The EISSMM model was empirically tested and confirmed at two universities, examining the success of the IS used in the teaching process to substantiate the proposed theoretical model rather than comparing the results from these two institutions. Based on the hypothesized relations in the model (Figures 2 and 3), the stakeholders in the teaching process can identify the critical factors that contribute to the IS's success. Likewise, they can foresee potential problems and shortcomings arising from applying the IS in the teaching process and, guided by this, approach the necessary improvement measures. Additionally, the EISSMM model can be used as an instrument for comparative analysis of the previous and current state using the IS. In addition, the model can be applied for comparative analysis in one institution and for comparison with other institutions that use such an IS. Having said that, our future research would be utilizing the model in some of the suggested settings.

Challenged by the COVID-19 pandemic, many organizations were forced to change their strategies and orient their business toward the virtual. In this regard, the EISSMM can identify whether this transition was successful by implementing IS to support the work environment. In such a situation, the model gains even more importance because it assesses workforce agility, evidenced to be significant when such unforeseen circumstances occur. Changes caused by sudden and unpredictable conditions also affect transitions

in various institutions that provide specific services. For example, standard business processes were implemented through electronic services after transitioning from a physical work environment to a digital one. The IS success that provides these e-services can be measured by applying the EISSMM, representing another significant practical implication of this research. This research is limited to testing the relations among six factors from the UTAUT model and three workforce agility factors. However, verifying the proposed EISSMM model is still needed to confirm certain relations. An insight into the descriptive statistics of relations between the respected factors and the demographic characteristics of the respondents, which can explain such phenomena, enabled the authors to assume reasons for not confirming them.

However, more detailed research on unconfirmed relations was not additionally performed within the scope of this research. Therefore, according to the aforementioned limitation, future research aims to research individual factors to obtain a more precise and concrete answer to the identified deficiency. In addition, repeated studies using the same methodology in the future could confirm the assumptions about the reasons for not establishing certain links in the model and thus provide the key stakeholders of the teaching process with a valid basis for improvement. This research is also limited in terms of the sample. Namely, data were collected from the UNS and the PICB. If the respondents belonged to another university or organization type, there is a possibility that the results would differ. Therefore, to increase the validity of the results, further research should test this model in different contexts and organization types.

**Author Contributions:** Conceptualization, T.V. and D.S.; Methodology, T.V. and D.S.; Software, T.V.; Validation, T.V. and D.C.L.; Investigation, T.V.; Resources, D.S., R.D., Â.O. and D.P.; Data curation, R.D., Â.O. and D.P.; Writing—original draft, T.V. and D.C.L.; Writing—review & editing, D.S., D.C.L., R.D. and D.P.; Visualization, D.C.L.; Supervision, D.S. All authors have read and agreed to the published version of the manuscript.

**Funding:** This research received no external funding.

**Institutional Review Board Statement:** Not applicable.

**Informed Consent Statement:** Not applicable.

**Data Availability Statement:** Not applicable.

**Conflicts of Interest:** The authors declare no conflict of interest.

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
