# Peer review of "The Extended Information Systems Success Measurement Model: e-Learning Perspective"

_applsci, doi:10.3390/app13053258_

Round 1

Reviewer 1 Report

The presented article deals with an interesting topic. The introduction and theoretical basis is processed at the required level. I consider up to 13 hypotheses to be a serious weakness of the article, which is an excessive amount for one research article and makes it difficult to follow what the research is looking at. This fact was also reflected in the discussion, which I consider to be written too briefly and is rather a summary of the Results section. I lack information in the discussion, which is the own benefit of the presented research. I recommend the authors to reduce the number of hypotheses and highlight the basic focus of the research, while at the same time pointing out their contribution to the current state of knowledge in the discussion. I think that the discussion does not serve to establish the authors how many hypotheses were confirmed and how many were not....

Reviewer 2 Report

The manuscript entitled "The Extended Information Systems Success Measurement Model"  investigates the crucial factors for measuring the success of the information system used in the e-learning process, considering the transformations in the work environment. 

Just as the authors mention, the topic approached by the authors is just as diverse as there are studies. Moreover, this topic is not an actual one, as a result less had been written about Information Systems Measurement Models in last years. As a consequence, the author have no single reference from 2022 and only 5 from 2021 (from a total of 129 titles). If I am mistaken, please update the literature review to as recently as possible.

Moreover, if there are already so many models, why do we need another one? What are the shortages of the previous ones and what is the insight offered by this new model? From lines 86-97 the innovative contribution of this research is not clear.

As an MIS and AIS professor and researcher, I was enthusiastic to receive the review just by the title. After reading the content, my enthusiasm decreased. Firstly, the title might be misleading as it was in my case: the manuscript is about IS in the e-learning context. I appreciate authors effort especially that two universities from different countries collaborated in this project. In my opinion, the problem is the topic itself, namely the lack of attractiveness while the effort devoted is evidently high.

Since the authors related the topic with the Covid-19 pandemic, I consider  the manuscript is more actual and I will present below some suggestions for improvement.

Minor issues:

- the title is too broad, not to say incomplete; it should specify that is about the IS used in the e-learning process; (in the meantime I received an updated file so I guess this issue is solved);

- Tables 4,5 don't have the legend for power "a" values. In SPSS, power "a" values are reporting errors or data inconsistency. Please elaborate on this aspect.

Major issues:

- the different number of responses (403 vs 161) from the two Universities and percentages (53,6% vs 45,5%) + two countries with different culture. This does not make the results comparable and it was obvious ( without all this efforts) that some results will differ ; the authors should explain how they perceive the comparability of their data and where lies the advantage of using the responds from those two Universities;

- in Table 2, for user behavior there is a single Manifest variable while for the other Factors there are at least four. This is an exception and should be explained otherwise might be perceived as an inconsistency in the model;

- the authors state in Abstract that "The study was motivated by the changes caused by COVID-19 and ..." but there is no reference to Covid-19 in the sections results, discussion or conclusion. 

I am sure the authors will deal with the above comments as they are experienced researchers and they have a good material. 

Round 2

Reviewer 1 Report

The authors have removed all the flaws. I recommend publishing the article.

Reviewer 2 Report

The authors dealt with each of my suggestions and thus the manuscript is improved. 

Congratulations to the authors and keep active the collaboration between the two universities!